# Distinct neurocomputational mechanisms support informational and socially normative conformity

Ali Mahmoodi[1,2,3]*, Hamed Nili[4,5], Dan Bang[6,7☯], Carsten Mehring[1,2☯], Bahador Bahrami[8,9,10☯]*

**1** Bernstein Centre Freiburg, University of Freiburg, Freiburg, Germany, **2** Faculty of Biology, University of Freiburg, Freiburg, Germany, **3** Wellcome Centre for Integrative Neuroimaging, Department of Experimental Psychology, University of Oxford, Oxford, United Kingdom, **4** Wellcome Centre for Integrative Neuroimaging, Centre for Functional Magnetic Resonance Imaging of the Brain, University of Oxford, Oxford, United Kingdom, **5** Department of Excellence for Neural Information Processing, Center for Molecular Neurobiology (ZMNH), University Medical Center Hamburg-Eppendorf (UKE), Hamburg, Germany, **6** Wellcome Centre for Human Neuroimaging, University College London, London, United Kingdom, **7** Department of Experimental Psychology, University of Oxford, Oxford, United Kingdom, **8** Faculty of Psychology and Educational Sciences, Ludwig Maximilian University, Munich, Germany, **9** Department of Psychology, Royal Holloway, University of London, Egham, United Kingdom, **10** Center for Adaptive Rationality, Max Planck Institute for Human Development, Berlin, Germany

☯ These authors contributed equally to this work.
* ali.mahmoodi1367@gmail.com (AM); bbahrami@gmail.com (BB)

**Data Availability Statement:** All relevant data are available at https://github.com/alimahmoodia/Reciprocity_Data/tree/main.

## Abstract

A change of mind in response to social influence could be driven by *informational* conformity to increase accuracy, or by *normative* conformity to comply with social norms such as reciprocity. Disentangling the behavioural, cognitive, and neurobiological underpinnings of informational and normative conformity have proven elusive. Here, participants underwent fMRI while performing a perceptual task that involved both advice-taking and advice-giving to human and computer partners. The concurrent inclusion of 2 different social roles and 2 different social partners revealed distinct behavioural and neural markers for informational and normative conformity. Dorsal anterior cingulate cortex (dACC) BOLD response tracked informational conformity towards both human and computer but tracked normative conformity only when interacting with humans. A network of brain areas (dorsomedial prefrontal cortex (dmPFC) and temporoparietal junction (TPJ)) that tracked normative conformity increased their functional coupling with the dACC when interacting with humans. These findings enable differentiating the neural mechanisms by which different types of conformity shape social changes of mind.

## Introduction

We are often faced with opinions that are different from our own. In these situations, we sometimes decide to stick to our own opinion and other times we change our mind. One key factor to select between these opposite social behaviours is our sense of confidence: The lower the confidence in our initial opinion, the higher the probability that we change our mind [1,2].

**Funding:** A.M. was supported by a PhD scholarship from the Graduate School Scholarship Program of the German Academic Exchange Service (DAAD). D.B. was supported by a Sir Henry Wellcome Postdoctoral Fellowship funded by the Wellcome Trust (213630/Z/18/Z). B.B. was supported by the Humboldt Foundation, the NOMIS Foundation, and the European Research Council under the European Union's Horizon 2020 research and innovation programme (grant agreement No. 819040 - acronym: rid-O). The funders had no role in study design, data collection and analysis, decision to publish, or preparation of the manuscript.

**Competing interests:** The authors have declared that no competing interests exist.

**Abbreviations:** dACC, dorsal anterior cingulate cortex; DDM, drift diffusion modelling; dmPFC, dorsomedial prefrontal cortex; GLM, general linear model; PPI, psychophysiological interaction; ROI, region of interest; TPJ, temporoparietal junction.

However, the way in which we process differing opinions has been shown to be influenced by a range of factors some of which are unrelated to accuracy, such as a desire to fit in with a group [3] or how receptive others have previously been towards us [4]. How people balance these epistemic and social factors remains an open and fundamental question in social cognitive neuroscience. Here, we develop an empirical framework for understanding the mechanisms that underpin social changes of mind at the cognitive and neural level.

There has been a recent interest in understanding changes of mind in nonsocial situations [1,5–7]. In a typical experiment, people are given the option to change their mind about an initial decision after being presented with additional (postdecision) evidence. People have been shown to solve this problem by computing the probability that the initial decision was correct given all the evidence, and markers of neural activity obtained with fMRI have revealed that this confidence computation is supported by dorsal anterior cingulate cortex (dACC) [1]. Interestingly, dACC also appears to play a central role in changes of mind in social situations [8]. For example, an initial set of fMRI studies focused on situations where people changed their subjective preferences (e.g., facial attractiveness ratings) after observing those of others [9–11]. Activity in dACC was found to track the observed difference between one's own and others' preferences and in turn predict whether people changed their reported preferences to align with those of others [9–11]. A more recent fMRI study asked people to make a perceptual decision after observing the recommendation of an advisor [12] and found that dACC activity tracked whether or not people based their decision on the advisor's response.

Traditionally, social influence—and thereby the factors that enter into social changes of mind—has been classified as informational or normative [13]. Informational influence is when we change our beliefs towards those of others in order to maximise accuracy. As in nonsocial situations, this process is likely to be governed by our sense of confidence in our own initial beliefs. A recent study [14] demonstrated that confidence is indeed a very reliable indicator of informational social influence. When people were more confident in their private decision, they sought less social information and were keen to persuade others to follow them.

By contrast, normative influence is when we change our beliefs towards those of others for reasons that are unrelated to accuracy. For example, we may seek to maximise group cohesion or social acceptance [3]. The challenge is that, while informational and normative factors are often in direct competition, they may nevertheless drive similar behavioural responses. For example, in the fMRI studies on subjective preference, a common interpretation is that people adapted their reported preferences towards others because they felt a pressure to conform to the group. However, people may have been uncertain about their own preferences [15] and used others as a cue to infer what they themselves feel [16]. Similarly, in studies where people made a perceptual decision after observing the recommendation of an advisor, people may have based their decision on the advisor's response because they felt that it was the socially right thing to do or because they genuinely had low confidence in their own sensory percept. It therefore remains an open question how the brain balances informational and normative factors during social changes of mind.

In the current study, we investigated the cognitive and neural basis of social changes of mind, by using a social perceptual decision task that directly separates informational and normative factors. On each trial, participants first made a perceptual estimate and reported their confidence in this response and were then presented with a partner's perceptual estimate. On one-half of trials, participants had the opportunity to revise their estimate. On the other half, the partner did. Unbeknownst to participants, we manipulated the degree to which the partner's revised estimate was influenced by the participant's initial estimate. Critically, this manipulation and concurrent sending and receiving social influence unveils a normative reciprocity effect: Participants are influenced more by the partner who has, in turn, been

influenced more by them, regardless of task performance [4]. In this way, one can separately measure the contribution of informational factors (the degree to which participants feel that they could improve on their perceptual estimate by taking into account that of the adviser, which is indexed by participants' confidence in their initial estimate) and normative factors (the degree to which participants feel that they should reciprocate influence, which is controlled by our manipulation of partner behaviour) to social changes of mind.

To anticipate our results, behavioural analyses revealed that participants' revision of their perceptual estimate was governed by both informational and normative factors: They shifted more towards the partner's estimate when they had low confidence in their own estimate and when there was a higher demand for reciprocating influence. Critically, in a control condition, where participants were told that they interacted with a computer, changes of mind were only influenced by informational factors. Analysis of fMRI data showed that dACC activity tracked both confidence and the demand for reciprocating influence at the time of revision. In line with the behavioural results, dACC activity tracked the normative factor only when participants believed that they worked with a human (but not a computer) partner. Further, we found that traditionally social brain areas—dorsomedial prefrontal cortex (dmPFC) and temporoparietal junction (TPJ)—tracked the degree to which a partner took into account the participants' perceptual estimate on trials where the partner revised their estimate and in turn increased its coupling with dACC on trials where the participants revised their estimate when both informational and normative demands were high. Taken together, these results support a general role for dACC in coordinating changes of mind in both nonsocial and social situations.

## Results

### Experimental task

Participants ($N$ = 60) performed a social perceptual decision-making task. In each experimental session, 3 participants came to the lab at the same time. The 3 participants met briefly and had their individual photos taken by the experimenter. One of the 3 participants performed the task while undergoing fMRI ($N$ = 20), and the remaining 2 participants performed the task in separate behavioural testing booths ($N$ = 40). The task consisted of 4 blocks of trials (scan runs)—with each participant paired with a unique partner in each block. Participants were told that in 2 of the 4 blocks, the partner was a computer. In each of the other 2 blocks, the participant was paired with one of the other participants. In reality, unbeknownst to the participants, all 4 partners were simulated. To help participants separate the partners, and to strengthen the computer-human distinction, participants were shown the photo of the current partner at the beginning of each trial.

In each trial, participants privately made a perceptual estimate about the location of a visual target and then rated their confidence in this estimate on a scale from 1 to 6 (Fig 1A). After having indicated their estimate and confidence, participants saw the partner's estimate of the location of the same visual target. The partner's estimate was generated by drawing a random sample from a Von Mises distribution centred on the correct answer. On odd (observation) trials, participants waited while the partner revised their estimate in light of the participants' estimate and were then shown the partner's revised estimate. On even (revision) trials, participants had the opportunity to revise their estimate in light of the partner's estimate, after which the revised estimate was shared with the partner. This arrangement gave the impression to the participants that in both types of trials, the revised estimate was shown to both players. To ensure that participants paid attention to the partner's estimate, they were required to place their revised estimate between their initial estimate and that of the partner. Participants did not receive feedback.

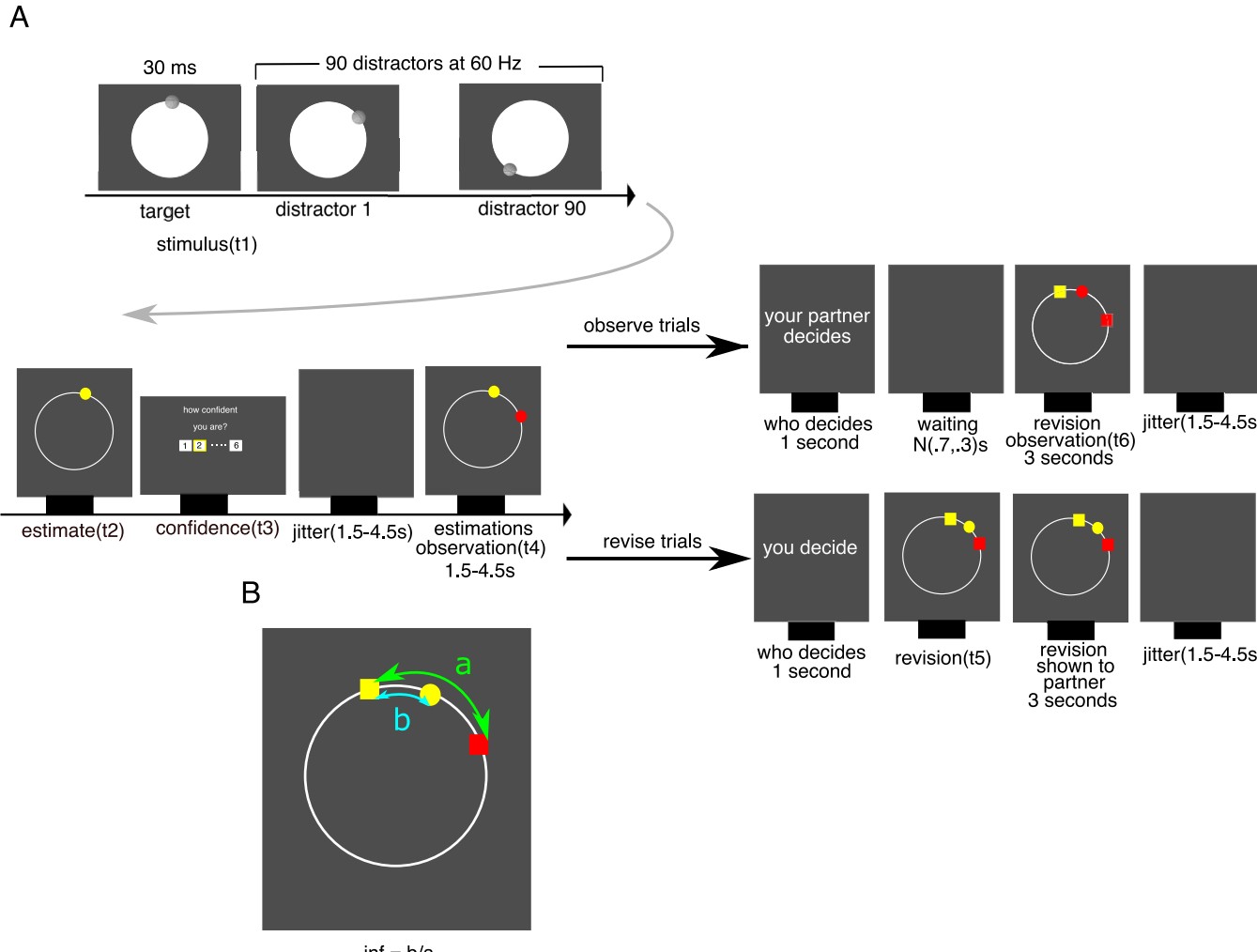

**Fig 1. Multistage decision-making framework for studying social changes of mind.** (**A**) On each trial, participants observed a sequentially presented series of dots on the screen (t1). They were then required to indicate where the very first dot in the series appeared (yellow dot; t2) and report their confidence on a discrete scale from 1 to 6 (t3). Next, they were presented with the estimate of a partner (red dot; t4) concerning the same stimulus. On half of the trials (observe), the partner had the opportunity to change their mind, and participants only observed the partner's revision. On the other half of trials (revise), participants had the opportunity to revise their initial estimate (t5). Participants were informed whether they were paired with computer or a human in each block. (**B**) Angular distance between initial estimates = a; angular distance between estimates after revision = b; social influence = b/a.

We manipulated the influence that participants exerted over their partners on observation (odd) trials: 1 human and 1 computer partner were strongly influenced by participants' estimate (susceptible blocks), whereas 1 human and 1 computer partner were only slightly influenced by participants' estimate (insusceptible blocks). In this way, the observation (odd) trials allowed us to introduce a normative aspect to the task—the degree to which the partner shifted towards the estimate made by the participant—whereas the revision (even) trials allowed us to quantify the impact of informational (confidence) and normative (influence) factors on social changes of mind.

## Behavioural separation of informational and normative factors

To disentangle the contribution of informational and normative factors to social changes of mind, we performed a linear mixed-effects regression analysis. All behavioural data analyses

include all behavioural and fMRI participants ($N = 60$). We predicted that, in revision trials, the degree to which participants changed their estimate towards that of the partner (from here onwards, *revision*) depends on the participant's *confidence* in their initial estimate, the influence that they exerted over their partner on the previous trial (i.e., *influence*) and the interaction between confidence and influence. In the interest of clarity, we define revision and influence separately here (see Fig 1B). Revision is defined by the angular difference between participant's initial and final estimates divided by the angular difference between the 2 initial estimates. Influence is defined by the angular difference between the partner's initial and final estimates divided by the angular difference between their 2 initial estimates. In agreement with a recent study [14], informational conformity would be demonstrated by a negative correlation between confidence and revision recorded in the same trial. A positive correlation between revision in the current trial and influence in the previous trial would be evidence for normative conformity. Critically, we would only expect this latter relationship to be observed for the human partner. This is because normative influence should, by definition, not pertain to human–computer interactions [4]. Therefore, for the human condition, we predict a positive effect of influence on revision and a potential negative effect of the interaction between confidence and influence as the effect of one might depend on the other.

Our first model (LLM1) also included a term for the partner type (human or computer) and interactions between partner type and our 3 variables of interest to directly test whether the effect of influence on revision is different between the 2 conditions. If this is the case, we expect a significant effect of the interaction between condition and influence on revision. We found that confidence had a negative effect on revision (parameter estimate: −0.33, 95% CI: [−0.42 −0.25], $F(1,93) = 58$, $p < 0.0001$), whereas influence had a positive effect (parameter estimate 0.11, 95% CI [0.03 0.18], $F(1,3475) = 9$, $p = 0.004$). Critically, there was an interaction between partner type and influence (parameter estimate: −0.09, 95% CI: [−0.16 −0.01], $F(1,3470) = 5.98$, $p = 0.01$)—indicating that the effect of influence was different for human and computer partners.

To unpack these results, we ran separate models for each partner type (LMM22). For the computer partner, there was a negative effect of confidence on revision (parameter estimate: −0.37, 95% CI: [−0.48 −0.26], $F(106,1) = 36$, $p < 0.001$) but no effect of influence (parameter estimate: 0.01, 95% CI: [−0.1 0.14], $F(408,1) = 0.16$, $p = 0.68$) and no interaction between confidence and influence (parameter estimate −0.03, 95% CI [−0.21 0.14], $F(1645,1) = 0.19$, $p = 0.65$) (Fig 2A). For the human partner, confidence had a negative effect on revision (parameter estimate −0.3, 95% CI [−0.41 −0.2], $F(127,1) = 34$, $p < 0.001$) and—in line with normative concerns being specific to human–human interactions—there was a positive effect of influence (parameter estimate 0.17, 95% CI [0.05 0.28], $F(407,1) = 9.65$, $p = 0.006$) and a negative effect of the interaction between confidence and influence (parameter estimate −0.21, 95% CI [−0.4 −0.03], $F(1085,1) = 6.41$, $p = 0.01$) (Fig 2A)—in other words, the higher the confidence, the lower the effect of influence on revision, and vice versa. We also included influence that the participants had over their partners in the last 4 trials, but none of the influences that participants exerted over their partners in the earlier trials had any statistically significant effect on revision. These results suggest that participants' revision towards their partner is biggest when confidence was low, and influence was high. Conversely, it suggests that the revision was lowest when confidence was high, and influence was low. To visualise this effect, we plotted revision as a function of confidence and influence. Confidence and influence were divided into small [0, 0.33], medium [0.33, 0.66], and high [0.66, 1] categories (Fig 2B). The result exactly matches our expectation: Revision was highest (lowest) when confidence was low (high) and influence was high (low). Another implication of our modelling results is that in the human condition, participants should have been more influenced by their susceptible

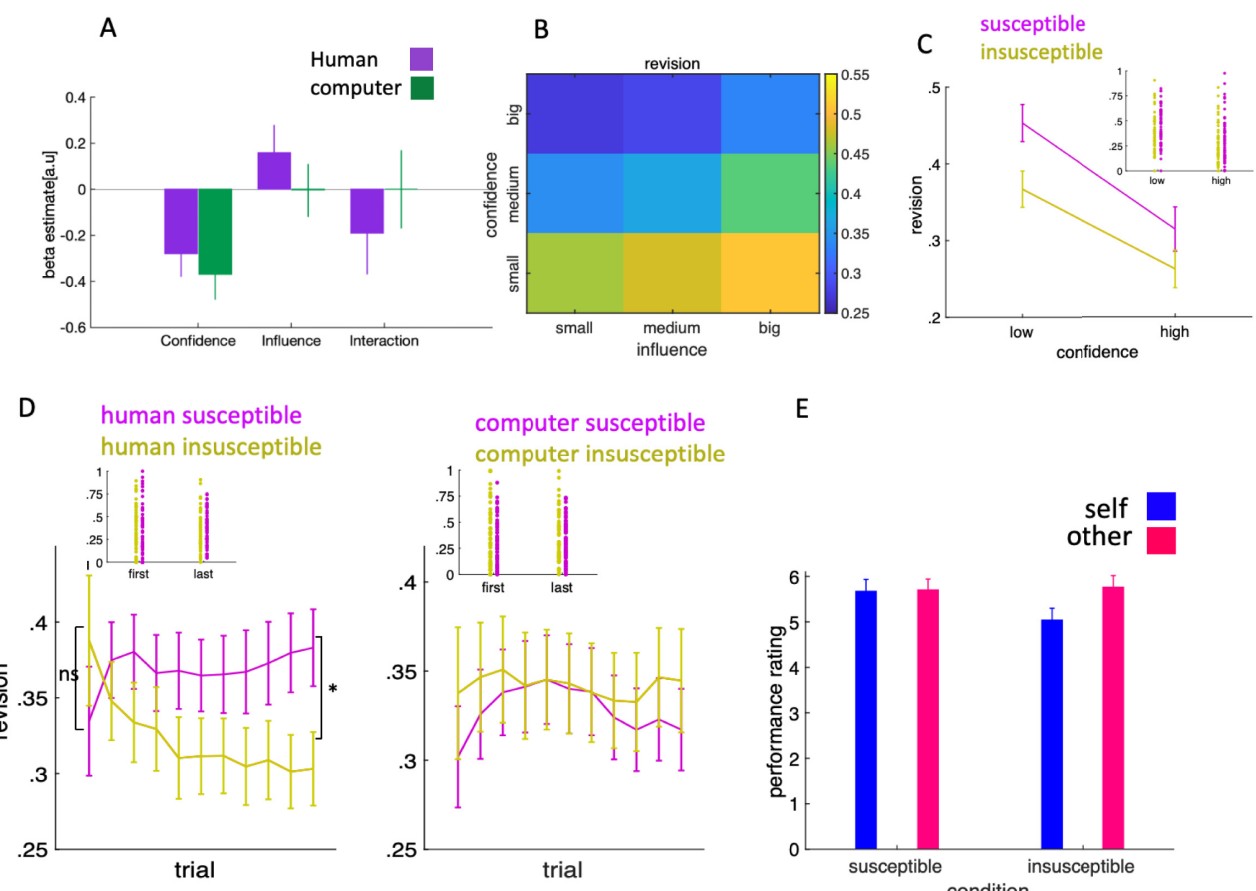

**Fig 2. Behavioural separation of informational and normative factors.** (**A**) We ran linear mixed-effects models separately for human and computer conditions in which we predicted participants' revision towards their partner using confidence, influence, and their interaction. Data are represented as group mean ± 95% confidence intervals. (**B**) Revision is plotted as a function of confidence and influence. This visualisation of our data indicates that revision was at its nadir when confidence was high, and influence was low. Participants' revision was at its zenith when confidence was low, and influence was high. (**C**) As predicted by the result of our linear mixed effect model, due to the positive effect of influence on revision, in the human condition, participants' revision towards their susceptible partner was significantly higher than their revision towards their insusceptible partner. (**D**) Revision is plotted separately for each condition. Participants increased (decreased) their revision towards their susceptible (insusceptible) human partner as they progressed in the experiment (left panel), while their revision towards their computer partner (either susceptible or insusceptible) remained unchanged in the computer condition. (**E**) Participants estimated their performance significantly lower than their partner in the insusceptible condition. In addition, their estimation of their own performance was significantly lower in the insusceptible condition compared to susceptible condition. Data and codes to recreate the figure are available at https://github.com/alimahmoodia/Reciprocity_Data/tree/main.

partner compared to their insusceptible partner. Consistent with the result of our linear model, in the human condition, we found that participants were on average more influenced by their susceptible partner compared to their insusceptible partner for both high and low confidences and this difference reached significant when we aggregated the trials across both high and low confidence levels (Wilcoxon sign-rank test W = 1,316, *p* = 0.003, Fig 2C).

The effect of influence on revision might be interpreted in 3 different ways. If participants only exercised positive reciprocity, then we expect that in the human condition, participants increased their revision towards their susceptible partner but did not change their revision towards their insusceptible partner during the experiment. However, if they exercised negative reciprocity, we expect that participants increased their revision towards the insusceptible partner but their revision towards their suspectable partner remained unchanged during the experiment. However, if they exercised both types of reciprocity, then we expect that the participants increased their revision towards their susceptible partner and decreased their

revision towards their insusceptible partner. To unveil the nature of reciprocity and its dynamic in our task, we computed the average revision in a sliding window of 5 trials, which moved with a step of 1 trial. We carried out this procedure for each condition separately (Fig 2D). As expected, in the human condition, when comparing very early trials, there was no difference between the magnitude of revision towards susceptible and insusceptible partners (Wilcoxon sign-rank test W = 598, $p$ = 0.4, Fig 2D), while revision towards their susceptible partner was significantly higher than their insusceptible partner in the last trials (Wilcoxon sign-rank test W = 1,302, $p$ = 0.008, Fig 2D). Comparing their revision towards their susceptible partner in the early and the last trials indicated that participants exercised positive reciprocity by increasing their revision towards their susceptible partner (Wilcoxon sign-rank test W = 643, $p$ = 0.04). Repeating the same analysis for the insusceptible condition revealed that participants significantly decreased their revision towards their partner during the experiment (Wilcoxon sign-rank test W = 1,211, $p$ = 0.03). Therefore, participants changed their revision towards either their susceptible or inscrutable partner by gravitating more towards the former and resist being influence by the latter. None of this behaviour was observed when we repeated this analysis for the computer partners.

The difference in revision between susceptible and insusceptible partners in the human condition could not be explained by any difference in perceived performance of self or partner between blocks. First, when playing with the human partners, participants' error was not different between the susceptible and insusceptible conditions (susceptible mean: 56° insusceptible mean 57.5°, Wilcoxon sign-rank test W = 1,071, $p$ = 0.25). At the end of each block, participants estimated their own and their partner's performance in the block on a scale ranging from 1 to 10. Participants' perception of their performance in insusceptible blocks was significantly lower than in the insusceptible blocks (Wilcoxon sign-rank test W = 678, $p$ = 0.02, Fig 2E). Comparing their estimate of their performance against their estimate of their partner's performance indicated that in the insusceptible block participants estimated their performance significantly lower than their partner (Wilcoxon sign-rank test W = 326, $p$ = 0.02, Fig 2E), while their estimate of their own performance was not significantly different from their estimate of their partners' performance in the susceptible blocks (Wilcoxon sign-rank test W = 483, $p$ = 0.68). We compared participants' estimate of the susceptible and insusceptible partners. Our data does not support any difference in estimated performance between susceptible and insusceptible partners (Wilcoxon sign-rank test W = 563, $p$ = 0.6), consistent with what we expected as we used the same algorithm to generate different partners' first choice.

And finally, the average revision value was 0.34, indicating that participants overall gave more weight to their own initial estimate (Wilcoxon sign-rank test against 0.5, median = 0.35, W = 582, $p < 0.0001$)—consistent with the finding that, all else being equal, people tend to discount the opinion of others [17].

## Encoding of informational and normative factors in dACC

Having established behaviourally that our experimental task dissociated the influence of informational and normative factors on social change of mind, we next turned to the fMRI data to identify neural substrates that may support the integration of these factors into a social change of mind. We focused our analyses on dACC as this area has consistently been linked to changes of mind in social [8] and nonsocial situations [1]. Our key question was whether dACC tracks both informational and normative factors, or only one of these factors, during social changes of mind. We used the same dACC region of interest (ROI) as in the study by Fleming and colleagues (2018) while noting that this ROI overlaps with the dACC clusters identified in the social studies discussed in the Introduction.

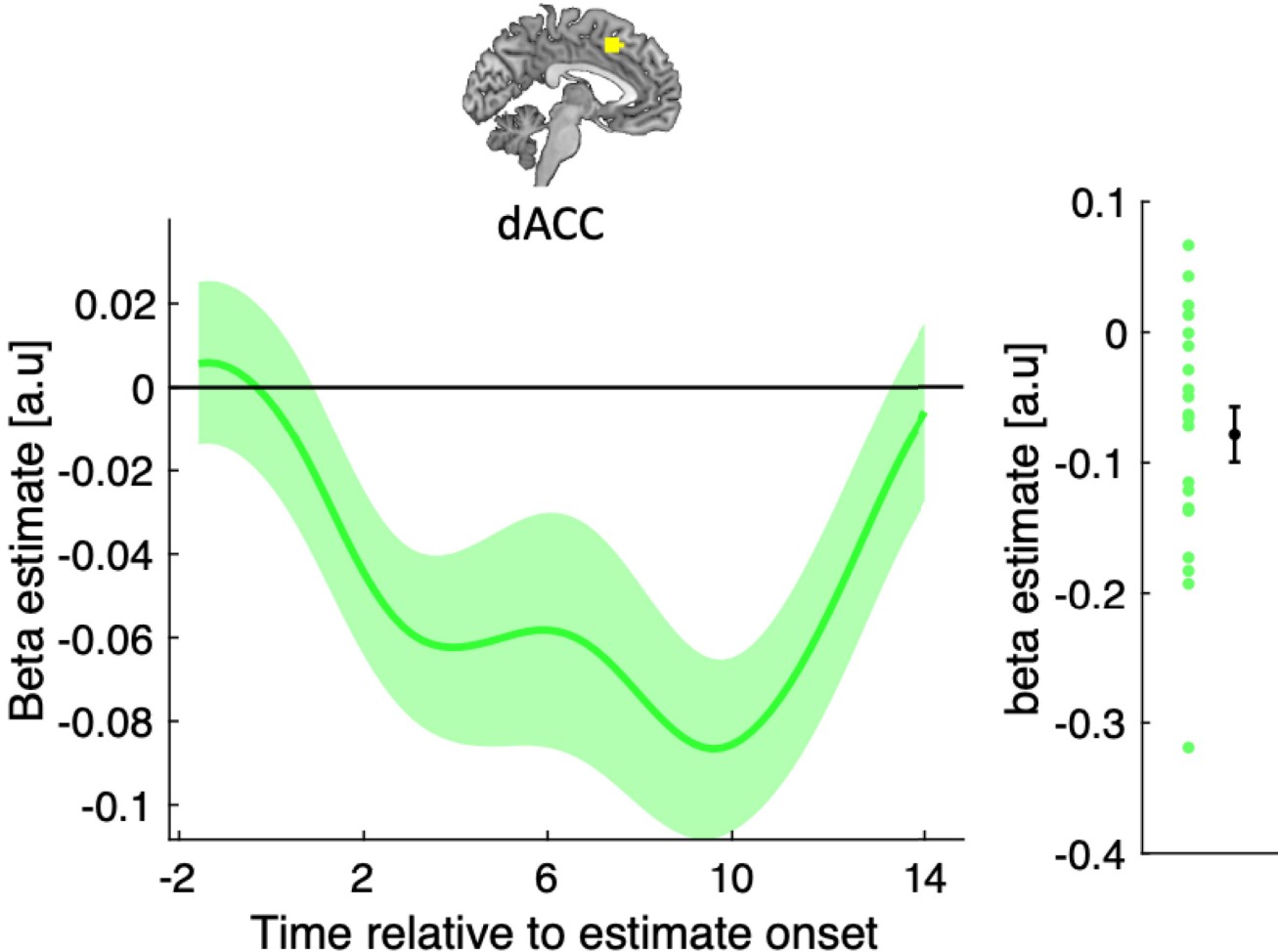

**Fig 3. Validation of the reported relationship between dACC activity and decision confidence.** GLM analysis of the effect of reported confidence on dACC activity time courses locked to the onset of the perceptual estimation screen. The group level significance was estimated using a leave-one-out procedure. Data are represented as group mean ± SEM. Vertical dashed line indicates estimation onset (t2). Data and codes to recreate the figure are available at https://github.com/alimahmoodia/Reciprocity_Data/tree/main. dACC, dorsal anterior cingulate cortex; GLM, general linear model.

We first asked whether dACC encodes participants' reported confidence at the time of the initial estimate (t2 in Fig 1). As shown by the behavioural results, confidence is a central component of informational conformity as it indexes the degree to which participants feel that they can improve on their perceptual estimate by considering that of a partner—regardless of whether the partner is human or a computer. General linear model (GLM) analysis of activity time courses locked to the onset of the estimation screen showed that dACC tracked confidence negatively on both human and computer trials (aggregated across both condition, Wilcoxon sign-rank test, $p = 0.002$, W = 24) (Fig 3)—a result that is consistent with the literature on the neural basis of confidence in nonsocial and social settings [18,19].

We next asked whether dACC encodes the normative component of social changes of mind —the demand to reciprocate social influence—when the participant revises their initial estimate. Additionally, we asked whether dACC continues to encode decision confidence at this stage. To this end, we performed a GLM analysis of activity time courses locked to the onset of the revision screen (t5 in Fig 1) using confidence, influence, and their interaction as predictors. Because the behavioural results showed that influence (its main effect or interaction with

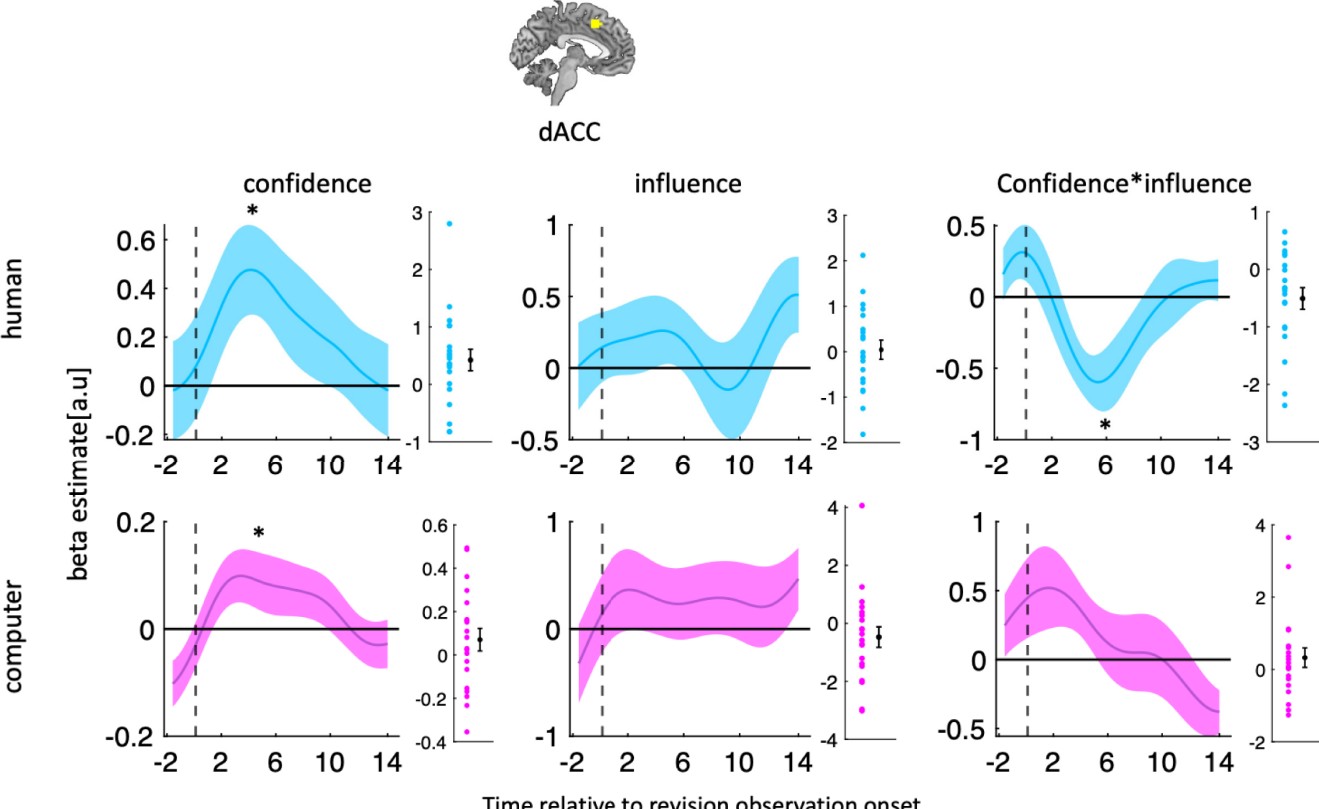

**Fig 4. dACC tracks informational and normative factors during social changes of mind.** GLM analysis of the effects of reported confidence, social influence, and their interaction on dACC activity time courses locked to the onset of the revision screen for the human (top) and the computer (bottom) condition. The star indicates that the time course was significantly different from zero using a leave-one-out procedure. Data are represented as group mean ± SEM. Vertical dashed line indicates revision onset (t5). The scatter plots show single subject estimate using leave-one-out procedure. Data and codes to recreate the figure are available at https://github.com/alimahmoodia/Reciprocity_Data/tree/main. dACC, dorsal anterior cingulate cortex; GLM, general linear model.

confidence) did not affect revision in the computer condition, we performed this analysis separately for the human and computer conditions. In line with the previous results, dACC tracked confidence at the time of revision in both the human and computer conditions aggregated across both conditions (aggregated across conditions, Wilcoxon sign-rank test, $p = 0.03$, W = 170) (Fig 4). Interestingly, dACC also tracked the interaction between confidence and influence at the time of revision but, critically, only in the human condition (Wilcoxon sign-rank test, $p = 0.04$, W = 46) (Fig 4). The interaction effect means that in the human condition, the response of the dACC to confidence was lower if influence was high and vice versa. Notably, we reached the same conclusion when we repeated this analysis using an alternative model (see S1 Text).

Next, we focused on the human condition and used the division of trials into revision and observation trials (Fig 1) to test whether the hypothesis that dACC tracked informational and normative factors specifically in the service of social changes of mind. If the dACC response pattern is driven by the prospect of having to make a second estimate, then we expect to find the encoding of informational and normative factors only in revision and not in observation trials. If, on the other hand, the task variables were automatically encoded regardless of current task requirements, then we would expect to see the dACC response in both trial types. To test this prediction, we used dACC activity time courses locked to the onset of the screen that

announced whether the current trial was a revision or an observation trial—notably, the screen appeared after participants had made their initial estimate and had been presented with that of their partner. In support of a specific role of dACC in revision, this analysis revealed that dACC did not track confidence, influence, or their interaction on observation trials (see S2 Text). Notably, we conducted Bayesian statistical analysis to show that the nonsignificant results that we reported in the dACC were not due to small sample size (see S3 Text). It might be argued that participants might have lowered attention on observation trials and in this in turn may have impacted the neural results. However, it should be noted that the fact that participants' influence on the partners as revealed by observation trials had an impact on participants' behaviour on revision trials shows that participants paid close attention to the task events. Because of the functional heterogeneity of the dACC [20–22], we chose a subregion of this area that was previously implicated in the change of mind [1]. To indicate that our results are not dependent on our specific ROI selection, we chose 3 more ROIs published in the literature [8,12] and found that all different ROIs lead to the same conclusion (see S4 Text).

## Encoding of normative factors in social brain areas

Having established that dACC integrates informational and normative factors during social changes of mind, we sought to identify the neural substrates that were most likely to provide the normative input to dACC. The informational component, i.e., confidence, is immediately available on a revision trial and may be encoded by dACC itself [1]. The normative component, i.e., social influence, however, must first be assessed on the preceding observation trial and then carried forward to the upcoming revision. We hypothesised that dmPFC and the TPJ may serve such assessor function. Both of these areas are part of the theory of mind network [23–25] and have been shown to track the trial-by-trial variation in task-relevant social variables [26,27] and social prediction [28]. To test this hypothesis, we first focused on observation trials and performed a GLM analysis of dmPFC and TPJ activity time courses locked to the observation of partner's change of mind (t6 in Fig 1) using the influence as a predictor. In line with our hypothesis, both dmPFC (Wilcoxon sign-rank test, $p = 0.002$, W = 189), and TPJ (Wilcoxon sign-rank test, $p = 0.01$, W = 173), tracked influence at the time of revision observation in the human condition (Fig 5A and 5B). Our TPJ and dmPFC masks were based on independent connectivity-based parcellations of the human brain [29,30]. As the social neuroscience literature is vast, this time—rather than identifying closely related studies—we created control masks using Neurosynth [31]. For both new masks, the TPJ and dmPFC responses to influence at the time of revision observation was similar to that seen for original ROIs and both set of responses were statistically significant ($p < 0.05$) (see S5 Text).

If dmPFC and/or TPJ provide the normative factor that is used by dACC to drive social changes of mind, then connectivity between these areas and dACC on revision trials should vary with the demand for reciprocating influence as computed on a preceding observation trial. To test this prediction, we performed a psychophysiological interaction (PPI) analysis in which we quantified connectivity between dmPFC/TPJ and dACC at the onset of the revision screen on human trials. This was done using a GLM where we predicted the activity of the dACC as a function of confidence, influence, TPJ activity, and all interaction terms. We included confidence, influence, and its interaction with confidence as the behavioural factors as our behavioural and neural results from the dACC show that the contribution of influence to social changes of mind depends on confidence, and vice versa (Figs 2 and 4). In support of our hypothesis, this analysis revealed a close coupling between TPJ and dACC. Shortly after the onset of the revision screen, TPJ–dACC connectivity varied with confidence (Wilcoxon

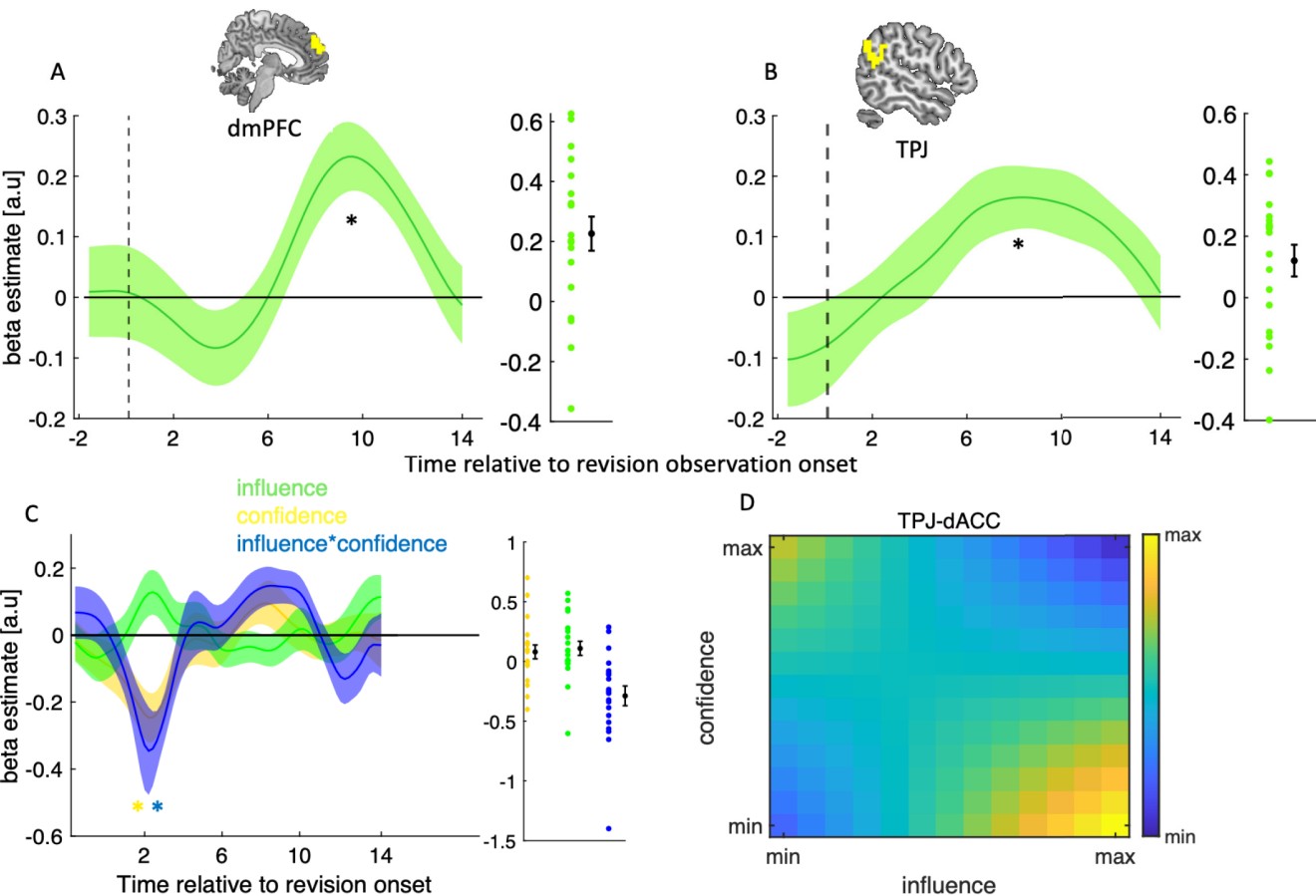

**Fig 5. Encoding of normative factors in social brain areas.** (**A, B**) dmPFC and TPJ tracks social influence on observation trials. GLM analysis of the effects of social influence on (left panel) dmPFC and (right panel) TPJ activity time courses locked to the onset of the revision observation screen. (**C**) PPI analysis of ROI activity time courses. Traces are coefficients from a GLM in which we predicted dACC activity from the interaction between TPJ activity and (1) confidence (yellow), (2) influence (green), and (3) the interaction between confidence and influence (blue)—while controlling for the main effect of each term (confidence, influence, and TPJ activity). (**D**) Visualisation of TPJ–dACC connectivity. Hotter colours indicate greater TPJ–dACC connectivity as a function of variation (in z-score units) in influence (x-axis) and confidence (y-axis). All values are z-scores and the difference in magnitude does not show any difference in the direction of the connectivity. TPJ–dACC connectivity was estimated using group-level coefficients averaged across a time window from 2 seconds to 3 seconds. In **A-B**, data are represented as group mean ± SEM. In A, B, anc C, the star indicates that the time course was significantly different from zero using a leave-one-out procedure. The scatter plots show single subject estimate using leave-one-out procedure. dACC, dorsal anterior cingulate cortex; dmPFC, dorsomedial prefrontal cortex; GLM, general linear model; PPI, psychophysiological interaction; ROI, region of interest; TPJ, temporoparietal junction.

sign-rank test, $p = 0.007$, W = 33), and its interaction with influence (Wilcoxon sign-rank test, $p = 0.002$, W = 25) (Fig 5C). Visualisation of these effects showed that TPJ–dACC connectivity was highest when influence was high, and confidence was low (Fig 5D)—the condition where normative factors have the largest influence on social changes of mind (Fig 2). There was also a peak when confidence was high and influence was low. We note that dmPFC–dACC connectivity showed the same pattern as TPJ–dACC connectivity but did not reach significance (see S6 Text). In addition, a direct comparison of the connectivity between human and computer conditions showed a significant difference between the 2 conditions (see S7 Text).

Finally, to complement our ROI analysis, we performed an exploratory whole-brain analysis in which we searched for neural correlates of our variables of interest. We confirm that we found the neural correlates of confidence but not influence and the interaction between confidence and influence at the whole brain level (See GLM1 in Methods and S8 Text).

## Discussion

A key feature of adaptive behavioural control is our ability to change our mind as new evidence comes to light. Previous research has identified dACC as a neural substrate for changes of mind in both nonsocial situations, such as when receiving additional evidence pertaining to a previously made decision [1], and social situations, such as when weighing up one's own decision against the recommendation of an advisor [12]. However, unlike the nonsocial case, the role of dACC in social changes of mind can be driven by different, and often competing, factors that are specific to the social nature of the interaction [13]. In particular, a social change of mind may be driven by a motivation to be correct, i.e., informational influence. Alternatively, a social change of mind may be driven by reasons unrelated to accuracy—such as social acceptance—a process called normative influence. To date, studies on the neural basis of social changes of mind have not disentangled these processes. It has therefore been unclear how the brain tracks and combines informational and normative factors [12,32,33].

Here, we leveraged a recently developed experimental framework that separates humans' trial-by-trial conformity into informational and normative components [4] to unpack the neural basis of social changes of mind. On each trial, participants first made a perceptual estimate and reported their confidence in it. In support of our task rationale, we found that, while participants' changes of mind were affected by confidence (i.e., informational) in both human and computer settings, they were only affected by the need to reciprocate influence (i.e., normative) specifically in the human–human setting. It should be noted that participants' perception of their partners' accuracy is also an important factor in social change of mind (we tend to change our mind towards the more accurate participants). Not being the focus of this study, we controlled for the effect of partners' accuracy by designing partners of equal performance. Participants' perceived performance of their partner was not different across conditions. In fact, they perceived their own performance as worse in the insusceptible (versus susceptible) block. These findings are not consistent with any account that explains our findings via a change of participants' evaluation of their partners' accuracy. In addition, participants assessed their own performance as worse than that of the insusceptible partner (Fig 2E). A purely informational account of conformity predicts that the participants should have revised more towards the insusceptible than the susceptible partner, but this was the opposite of what we observed (Fig 2E).

Building on previous research on the neural basis of changes of mind [1,10,12], our analysis of fMRI data acquired during task performance focused on dACC and, in particular, the degree to which dACC encoded informational and normative factors in the different conditions of our experiment. Overall, our findings support a central role for dACC in orchestrating social changes of mind. First, in line with the behavioural results, when participants were given the opportunity to revise their initial estimate, dACC tracked confidence both when the partner was a human or a computer. However, only in the human condition did dACC concurrently track the need to reciprocate influence. Moreover, in the human partner condition, dACC coding of confidence and reciprocity was restricted to when participants had the opportunity to revise their initial estimate (revision trials) but not when it was the partner's turn to do so (observation trials). Together, these findings demonstrate that the dACC responses were directly tied to social behavioural control. More broadly, looking beyond changes of mind, our neural results are in line with a proposal that all task-relevant variables, independent of their origin and nature, converge in dACC and that dACC in turn supports the selection of task-appropriate behavioural responses [20].

If dACC supports the integration of informational and normative factors into a social change of mind, which brain regions provide the respective inputs to this process? As for the

informational component of a social change of mind, our results suggest, at first glance, that dACC itself may be involved in the construction of decision confidence. In particular, we found that, at the time of making a perceptual estimate, dACC tracked participants' confidence in this estimate—a temporal association identified by other studies on the neural basis of decision confidence [19]. However, recent research, which disentangled the components of decision confidence [34] or separated the subjective sense of confidence from explicitly shared report of confidence [18], suggest that this temporal association is due to a role of dACC in controlling confidence-based behaviours rather than encoding a sense of confidence per se. We acknowledge that our whole-brain analysis did not reveal any other brain regions that may have supported a confidence computation. As for the normative component of a social change of mind, our analysis of TPJ and dmPFC—both implicated previously in the so-called theory of mind network [23,24]—suggests that these regions may provide the social context for a change of mind. First, on observation trials—i.e., when observing a human partner's revised estimate—TPJ and dmPFC tracked the degree to which a partner took into account the participant's estimate and thereby the degree to which participants should reciprocate influence on the subsequent revision trial. Second, on revision trials—when informational and normative considerations should be balanced against one another—functional coupling between dACC and TPJ was highest when the normative component of a social change of mind was required in the conformity process.

Our results prompt a reconsideration of earlier accounts of the role of dACC in social changes of mind. For example, Qi and colleagues (2018) found that dACC activity predicted the degree to which participants' perceptual judgements deviated from those of an advisor [12]. Invoking the conflict monitoring theory of dACC function [35], Qi and colleagues (2018) took this response pattern to suggest that dACC tracks social conflict. However, as highlighted by our study, decision confidence and social conflict are often two sides of the same coin—the higher the degree of confidence, the lower the influence of others—making it hard to arbitrate between a conflict account of dACC and its role in encoding informational and normative components of conformity. In order to test the social conflict account of dACC, we reran the GLM analysis of dACC activity time courses locked to the onset of the revision screen, including the difference between participants' initial estimate and the partner's estimate as well as the difference between participants' revised estimate and the partner's estimate in addition to confidence, the need to reciprocate and the interaction between confidence and the need to reciprocate. Notably, neither of the difference terms—both markers of social conflict as quantified by Qi and colleagues (2018)—were encoded by dACC. While our results do not rule out that dACC may track social conflict, they show that social conflict does not provide a unified explanation of dACC function during social changes of mind. Rather, dACC appears to track any variable—irrespective of whether it is informational or normative in nature—that are deemed relevant in the context of the current task at hand. It should be noted that a limitation of these analyses is given by the limited spatial resolution of fMRI. Hence, our analyses cannot reveal what is encoded in the activity of individual or small populations of neurons in dACC. Thus, our results cannot exclude that different neuronal subpopulation in dACC may contribute to encoding normative and informational aspects of conformity or that these signals are multiplexed within individual neurons or small populations.

Several studies have examined social influence broadly construed—including conformity, emulation, compliance, and imitation—across multiple domains. These domains include perceptual decision-making [14,36]; value-based decision-making [37,38]; object preference [10,39]; moral decision-making [40]; charitable giving [41]; and social punishment [42]. However, with the exception of one study [41], these studies did not include a nonsocial control similar to the computer condition in the current study. The exclusive use of human–human

interactions means that these studies cannot easily isolate the normative aspects of social influence. Our study goes beyond those earlier works in this respect and could pave the way for future studies of the behavioural and neural basis of different forms of social influence.

A number of the studies mentioned above used drift diffusion modelling (DDM) to assess how social and private information were combined into a decision [14,36,40,42]. Two studies [36,42] presented the social information before the participants had made their own private decision; one study [14] had the participants make their private decision first, and a final study [40] completely separated the private and social decision stages in an emulation paradigm that did not measure conformity. Remarkably, all studies found that social information changed the rate of evidence accumulation (drift rate). Our paradigm did not employ binary choice and therefore does not lend itself to a DDM analysis. However, the motivation of our design and our results are consistent with those of Tump and colleagues [14] in that confidence is the key factor determining informational conformity. Future research would benefit from combining a DDM approach to our human versus computer design.

Our results showed that humans can and do interact with nonhuman social partners via informational but not normative conformity. When humans interact with an inanimate computer partner, this form of normative conformity is not observed neither in behaviour nor in the human brain. This will have important ramifications for the new and burgeoning field of human–AI interactions. For example, with the imminent introduction of self-driving cars into everyday life, studies such as ours will be able to help anticipate the emergence of norms of politeness between human and AI drivers on the road. It is important to note that reciprocity is only one among many well-known social norms. When we interact with others, (among other motives) we also wish to be included and accepted, be treated fairly, and have our right to privacy, dignity, and agency respected [43]. In choosing to focus on reciprocity, we were motivated by its longstanding history and the widespread consensus on this norm among different cultures [44]. Future research should address the extent to which the same neurocomputational mechanisms support different social norms.

One limitation of our study is that we only studied the impact of 2 types of social influence on social changes of mind. However, various motives contribute to people's social behaviour [45]. Seeking elevated accuracy [46] is perhaps the most popularly recognised motive but only one among many other motives. Other motives include diffusion of responsibility and regret [47] and equality [2]. Future research could help determine whether and to what extent our participants were driven by these various motives.

Another limitation of our study is that our visual stimulation paradigm was not optimised for activating the early visual cortex. This design choice could explain why the whole-brain search did not identify social effects on sensory processing. A long-standing question in the literature of social conformity [48] is whether social information changes the brain processes at the level of sensation or at the level of cognition and decision-making. While numerous studies have sought to address this issue, the debate remains unresolved [10,49,50]. One avenue for future research is to combine our social paradigm with those that are known to drive visual activations in a reliable and retinotopically organised manner (e.g., high-contrast, rapidly flickering Gabor patches) to address this question.

Finally, we recognise that sample size ($N = 20$) for the number of participants in our fMRI experiment puts some limitation on conclusions that can be drawn from our findings. To address this issue, we used methods from Bayesian statistics to show that our negative findings (especially, the absence of a reciprocity effect in the computer condition) was not likely to be due to small sample size. We have made our data available to researchers interested in this paradigm and assessing normative conformity in future studies to help them plan their study sample size.

## Methods

### Participants

In total, 60 healthy adult participants (30 females, mean age ± std:25 ± 3) participated in the experiment after having given written informed consent. The experimental procedure was approved by the ethics committee at the University College London (UCL) (ethics ID 4223/002), and the study was conducted according to the principles expressed in the Declaration of Helsinki.

### Experimental paradigm

Participants were presented with a sequence of 91 visual stimuli consisting of small circular Gaussian blobs (r = 5 mm) in rapid serial visual presentation on the screen. The first stimulus was presented for 30 ms, while every other stimulus was presented for 15 ms each. Participants' task was to identify the location of the first stimulus. Participants were required to wait until the presentation of all stimuli were finished, and then indicate the location of the first stimulus using a keyboard. The reported location was marked by a yellow dot. After participants reported their initial estimate, they were required to report their confidence about their estimate on a numerical scale from 1 (low confidence) to 6 (high confidence). For participants in the fMRI scanner, this stage was followed by a blank jitter randomly drawn from a uniform distribution from 1.5 to 4.5 seconds. Afterwards, participants were shown the estimate of their partners about the same stimulus for 1.5 seconds by a small red dot on the screen (plus a jitter time randomly drown from a uniform distribution from 1.5 to 4.5 seconds for the fMRI experiment). Then, either the participant revised her estimate or observed the partner revise theirs. After the second estimate was made, all estimates were presented to the participants for 3 seconds (plus a jitter time randomly drawn from a uniform distribution from 1.5 to 4.5 seconds for the fMRI experiment). In this stage, the first estimate was shown by a hexagon (for participants outside the fMRI scanner) or by a dot with a different colour (for the fMRI experiment) to be distinguished from the second estimate, which was shown by a circle (Fig 1B). Participants were told that their payoff would be calculated based on the accuracy of their first and second estimates. However, everyone was given a fixed amount at the end of the experiment. In the fMRI experiment, 10 participants' dot colour was yellow and their partner's dot colour was red. For the remaining 10 participants, the colours were reversed. Further details of the experimental paradigm are described in our previous study [4].

Three participants came to the MRI facilities at the same time. After reading the task instructions, 1 participant was selected to perform the experiments in the fMRI scanner while the other 2 carried out the behavioural task outside the scanner. Participants were told that they will play with 4 different partners: 2 human partners (the two they met before the experiment) and 2 computer partners, which were controlled by the algorithm described below. Participants completed 4 blocks (scan runs) of the experiment each consisting of 30 trials. In each block, they only worked with 1 partner. At the beginning of each trial, a photo of the partner they work with was shown to the participants. Photos of 2 different computers with different colours (counterbalanced across participants) represented the 2 computer partners. In reality, and unknown to the participants, all partners' estimates were generated by a computer algorithm. The partners only differed in the way they generated their second choice. In the insusceptible blocks, participants' influence over their partner was chosen randomly from a uniform distribution on the interval [0, 0.2]. For the susceptible partner, participants' influence was chosen with a probability of 0.5 from a uniform distribution on the interval [0.7, 1], with a probability of 0.2 from a uniform distribution on the interval [0.3, 0.7], and with a probability of 0.3 from a uniform distribution on the interval [0, 0.3].

All experiments were performed using Psychophysics Toolbox [51] implemented in MATLAB (Mathworks). The behavioural data were analysed using MATLAB.

## Debriefing

After each session of the experiment, all participants were debriefed to assess to what extent they believed the cover story. We interviewed them with indirect questions about the cover story and all participants stated that they believed they were working with other human participants in neighbouring experimental rooms (if they were told that their partner is a human partner).

## Constructing partners

The estimates of partners were calculated similarly to our previous study [4]. In each trial, we drew the first choice of the partner from a von Mises distribution centred on the target with a concentration parameter kappa = 7.4 except in high confidence trials (confidence level of 5 or 6) where the partner's choice was randomly drawn from a uniform distribution centred on the participants' choice with a width of +/− 20 degrees in the behavioural experiment and +/− 50 degrees in the neuroimaging experiment. The same algorithm was used for generating human and computer partners.

Participants were simply told that they would interact either with a human being or a computer algorithm. Participants were not informed about the partners' accuracy or strategy in either the human or the computer blocks.

## Linear mixed effect model 1 (LMM1)

To investigate the difference between the experimental conditions on the effect of informational and normative factors on revision, we designed a model as follows:

$$
r_t = \beta_{1s} + \beta_{2s} \times c_t + \beta_3 \times inf_{t-1} + \beta_4 \times cond + \beta_5 \times c_t \times inf_{t-1} + \beta_6 \times c_t \times cond + \beta_7 \\ \times inf_{t-1} \times cond + \beta_8 \times c_t \times inf_{t-1} \times cond \tag{1}
$$

$r_t$ and $c_t$ correspond to the participants' revision and confidence on trial t, respectively. $inf_{t-1}$ corresponds to the influence that participants exerted on their partner on trial t-1. *Cond* corresponds to condition included as dummy variable (1 = human, 2 = computer). The intercept ($\beta_{1s}$) and only the slope associated with confidence was allowed to vary across participants by including random effects of the form $\beta_{ks} = \beta_{k0} + b_{ks}$ where $b_{ks} \sim N(0, \sigma_k^2)$.

The statistics for confidence, influence, and the interaction between influence and condition are reported in the main text. The statistics for the remaining regressors were as follows: condition (parameter estimate 0.7, 95% CI [−0.004 0.16], F(3464,1) = 4.1, $p$ = 0.04), confidence and influence interaction (parameter estimate −0.4, 95% CI [−0.82 0.01], F(3478,1) = 5.4, $p$ = 0.02), confidence and condition interaction (parameter estimate −0.1, 95% CI [−0.22 0.01], F(3468,1) = 3.02, $p$ = 0.08), and the triple interaction between confidence, influence, and condition (parameter estimate 0.21, 95% CI [−0.05 0.47], F(3472,1) = 3.65, $p$ = 0.05).

## Linear mixed effect model 2 (LMM2)

To investigate the distinct effect of informational and normative factors on revision, we designed a model as follows:

$$
r_t = \beta_{1s} + \beta_{2s} \times c_t + \beta_{3s} \times inf_{t-1} + \beta_4 \times c_t \times inf_{t-1} \tag{2}
$$

$r_t$ and $c_t$ correspond to the participants' revision and confidence on trial t, respectively. $inf_{t-1}$

corresponds to the influence that participants exerted on their partner on trial t-1. The intercept ($\beta_{1s}$) and all slopes ($\beta_{2s}, \beta_{3s}$) were allowed to vary across participants by including random effects of the form $\beta_{ks} = \beta_{k0} + bks$ where $b_{ks} \sim N(0, \sigma_k^2)$. This model was fitted separately to the data from human and computer conditions. $\beta_4$ was fixed across participants as the full model (where $\beta_4$ varies across participants was overparametrized and the Hessian matrix was not positive definite. We therefore defined $\beta_4$ as fixed effect. We note that this model has lower BIC (BIC = 745) than the models with confidence as fixed effect (BIC = 756) or influence as fixed effect (BIC = 753).

Notably, in both above models (LMM1 and LMM2) by dividing all confidence values by 6, we normalised confidence to lie between 0 and 1.

We extended our LMM2 by adding a setting variable (fMRI, Behavioural), to test for any difference between behavioural and fMRI participants. Critically, we did not find any difference between the interaction of any of the variables of interests and setting across conditions. We therefore did not include this term in the final analysis.

We have corrected for multiple comparison using Holm–Bonferroni correction in our linear mixed effect models and all neuroimaging analyses. For both models, we assessed the statistical significance of model parameters by F-statistics and Satterthwaite's approximation for degrees of freedom.

## MRI data acquisition

Structural and functional MRI data were obtained using a Siemens Avanto 1.5 T scanner equipped with a 32-channel head coil at the Birkbeck-UCL Centre for Neuroimaging. The echoplanar image sequence was acquired in an ascending manner, at an oblique angle ($\approx 30°$) to the AC–PC line to decrease the impact of a susceptibility artefact in the orbitofrontal cortex with the following acquisition parameters: Each volume contained 44 slices of 2 mm thickness, 1 mm slice gap; echo time = 50 ms; repetition time = 3,740 ms; flip angle = 90°; field of view = 192 mm; matrix size = 64 × 64. A structural image was obtained for each participant using MP-RAGE (TR = 2730 ms, TE = 3.57 ms, voxel size = 1 $mm^3$, 176 slices). Each scan run (4 in total) lasted on average 15 minutes (range: [12 to 17 minutes])–generating 120 trials of behavioural data and 176 to 251 brain volumes for fMRI data.

## fMRI data analysis

Imaging data were analysed using Matlab (R2016b) and Statistical Parametric Mapping software (SPM12; Wellcome Trust Centre for Neuroimaging, London, UK). Images were corrected for field inhomogeneity and corrected for head motion. They were subsequently realigned, coregistered, normalised to the Montreal Neurological Institute template, spatially smoothed (8 mm FWHM Gaussian kernel), and high filtered (128 seconds) following SPM12 standard preprocessing procedures.

The design matrix for GLM1 included 6 events. These were the times of stimulus representation (t1), making the first (private) estimate (t2), reporting the confidence (t3), showing the first estimates (t4), making the second (revised) estimate (t5), and revision observation (t6). Furthermore, regressors t2 and t3 were parametrically modulated by participant's reported confidence. The regressor for t5 included the parametric modulators confidence, angular distance between the participant's own and the partner's first estimate, angular distance between participant's second and the partner's first estimate, the amount of revision that participants made towards their partner's estimate, participants' influence over their partner in the previous trial, and the interaction between this influence and confidence. The regressor for t6 included the participants' influence over their partner as parametric modulator. For events in

which the duration depended on the participants' reaction time (t2, t3, and t5), the natural logarithm of the reaction time, i.e., log(RT) was included as the parametric modulator. Parametric modulators were not orthogonalized to allow the regressors to compete for explaining the variance.

## Regions of interest analysis

We focused on 3 a priori ROIs highlighted by previous research on social cognition. The TPJ mask was defined using the Human TPJ parcellation study developed by [30] and mirrored to the left hemisphere to create a bilateral mask. We used the dmPFC mask from [29] and the dACC mask from [1]. We transformed each ROI mask from MNI to native space and extracted preprocessed BOLD time courses as the average of voxels within the mask. For each scan run, we regressed out variation due to head motion, and upsampled the BOLD time course by a resolution of 0.2 seconds. For each trial, we extracted activity estimates in a 15-second window (75 time points), time-locked to 1 second before the onset of each event of interest. We used linear regression to predict the ROI activity time courses. More specifically, we applied a linear regression to each time point and then, by concatenating beta weights across time points, created a beta weight time course for each predictor of a regression model. We performed this step separately for each participant and pooled beta weight time courses across participants for visualisation. For example, for the dACC time course analysis for Fig 4, we used the following linear regression model:

$$dACC_{BOLD} = \beta_0 + \beta_1 \times c_t + \beta_2 \times inf_{t-1} + \beta_3 \times c_t \times inf_{t-1} + \beta_4 \times rt_t + \beta_5 \times r \qquad (3)$$

where $c_t$ correspond to the participants' confidence on trial t and $inf_{t-1}$ corresponds to the influence that participants exerted on their partner on trial t-1. $rt_t$ corresponds to participants' reaction time in trial t, and $r$ corresponds to block number.

We tested group-level significance using a leave-one-out procedure to avoid any selection bias. For each participant and for each time course signal, we computed the peak the signal (positive or negative) for the group and calculated the beta weight of the left-out participant at the time of the group peak. We repeated this procedure for each participant and compared the resulting beta weights against zero.

## Supporting information

**S1 Text. Alternative model for Fig 4 analysis in the main text.**
(DOCX)

**S2 Text. dACC activity on observation trials.** dACC, dorsal anterior cingulate cortex.
(DOCX)

**S3 Text. Bayesian statistical analysis of the effect of informational and normative conformity in the dACC.** dACC, dorsal anterior cingulate cortex.
(DOCX)

**S4 Text. Robustness of our dACC ROI analysis.** dACC, dorsal anterior cingulate cortex; ROI, region of interest.
(DOCX)

**S5 Text. Robustness of our dmPFC and TPJ ROI analysis.** dmPFC, dorsomedial prefrontal cortex; ROI, region of interest; TPJ, temporoparietal junction.
(DOCX)

**S6 Text. Connectivity analysis between dmPFC and dACC.** dACC, dorsal anterior cingulate cortex; dmPFC, dorsomedial prefrontal cortex.
(DOCX)

**S7 Text. PPI comparison between human and computer conditions.** PPI, psychophysiological interaction.
(DOCX)

**S8 Text. Exploratory whole-brain analysis.**
(DOCX)

## Author Contributions

**Conceptualization:** Ali Mahmoodi, Carsten Mehring, Bahador Bahrami.

**Data curation:** Ali Mahmoodi, Hamed Nili, Bahador Bahrami.

**Formal analysis:** Ali Mahmoodi, Hamed Nili, Dan Bang, Carsten Mehring, Bahador Bahrami.

**Funding acquisition:** Carsten Mehring, Bahador Bahrami.

**Investigation:** Ali Mahmoodi, Hamed Nili, Dan Bang, Carsten Mehring, Bahador Bahrami.

**Methodology:** Ali Mahmoodi, Hamed Nili, Dan Bang, Carsten Mehring, Bahador Bahrami.

**Project administration:** Ali Mahmoodi, Carsten Mehring, Bahador Bahrami.

**Resources:** Ali Mahmoodi, Carsten Mehring, Bahador Bahrami.

**Software:** Ali Mahmoodi, Hamed Nili.

**Supervision:** Hamed Nili, Dan Bang, Carsten Mehring, Bahador Bahrami.

**Validation:** Ali Mahmoodi, Carsten Mehring, Bahador Bahrami.

**Visualization:** Ali Mahmoodi, Carsten Mehring, Bahador Bahrami.

**Writing – original draft:** Ali Mahmoodi, Dan Bang, Bahador Bahrami.

**Writing – review & editing:** Ali Mahmoodi, Dan Bang, Carsten Mehring, Bahador Bahrami.

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
