## [Editor Report · Decision Letter 0]

13 Jul 2021

Dear Dr Mahmoodi, 

Thank you for submitting your manuscript entitled "Distinct neurocomputational mechanisms support informational and normative conformity" for consideration as a Research Article by PLOS Biology.

Your manuscript has now been evaluated by the PLOS Biology editorial staff, as well as by an academic editor with relevant expertise, and I am writing to let you know that we would like to send your submission out for external peer review. Please accept my apologies for the long delay in sending this decision to you.

Before we can send your manuscript to reviewers, we need you to complete your submission by providing the metadata that is required for full assessment. To this end, please login to Editorial Manager where you will find the paper in the 'Submissions Needing Revisions' folder on your homepage. Please click 'Revise Submission' from the Action Links and complete all additional questions in the submission questionnaire.

Please re-submit your manuscript within two working days, i.e. by Jul 15 2021 11:59PM.

Kind regards,

Gabriel Gasque

Senior Editor

PLOS Biology

ggasque@plos.org

---

## [Decision Letter · Decision Letter 1]

20 Aug 2021

Dear Dr Mahmoodi,

Thank you very much for submitting your manuscript "Distinct neurocomputational mechanisms support informational and normative conformity" for consideration as a Research Article at PLOS Biology. Your manuscript has been evaluated by the PLOS Biology editors, by an Academic Editor with relevant expertise, and by three independent reviewers. Please accept my apologies for the delay in sending the decision below to you.

The reviews of your manuscript are appended below. As you will see, all reviewers, and the Academic Editor as well, have expressed enthusiasm about your study. However, based on their specific comments and following discussion with the Academic Editor, I regret that we cannot accept the current version of the manuscript for publication. We remain interested in your study and we would be willing to consider resubmission of a comprehensively revised version that thoroughly addresses all the reviewers' comments. We cannot make any decision about publication until we have seen the revised manuscript and your response to the reviewers' comments. Your revised manuscript would be sent for further evaluation by the reviewers.

You will see that all reviewers agree that the number of participants in the fMRI experiment is concerningly low. In addition, the Academic Editor notes that the 20 subjects completed only 120 trials each (4 runs with 30 trials each) and it appears that only 44 fMRI volumes were collected per session, which would come out to about 3 minutes per run (12 minutes total). We think this is extremely little data for an imaging study. Therefore, we urge you to add more data to your study and increase the number of subjects to the standard in the field.

We appreciate that these requests represent a great deal of extra work, and we are willing to relax our standard revision time to allow you six months to revise your manuscript. 

**IMPORTANT - SUBMITTING YOUR REVISION**

*Resubmission Checklist*

*Published Peer Review*

*PLOS Data Policy*

*Blot and Gel Data Policy*

Sincerely,

Gabriel Gasque

Senior Editor

PLOS Biology

ggasque@plos.org

REVIEWS:

Reviewer #1: In this paper, the authors reported an fMRI study aiming to disentangling the neural substrates of "informational" vs. "normative" social conformity (or "social change of mind"). To this end, they developed a novel behavioral task by combining a visual perception task with social influence manipulation. The authors statistically dissociated informational and normative conformity by regressing the degree of behavioral changes on to participants' confidence of their own performance (i.e., informational conformity) and how much the influencer was influenced by the participants on separate trials (i.e., normative conformity). Neurally, the authors found that informational conformity was associated with dACC activity, regardless of whether the "information" was from a human or a computer agent; normative conformity also appeared to be correlated with dACC activity, but only when the confidence level was low and when the information was from a human agent. Normative information seems to be encoded in the social-called mentalizing network (e.g., dmPFC and TPJ) and conveyed to dACC via functional connectivity.

Overall, I enjoy reading this well-written paper. The behavioral task and statistical analysis offered a novel approach to quantify and dissociate informational and normative conformity (or social influence). The use of a computer agent condition was a clever way of demonstrating the social specificity of "normative" conformity. The fMRI analysis was guided by logically sound hypothesis and theorization. I have a few comments and suggestions.

1. The fMRI sample size (N = 20) is somewhat concerning. I urge the authors carry out post-hoc power analysis and report the achieved power of their various analysis (both behavioral and neural). This should be relatively straightforward because the fMRI results were primarily based on ROI analysis, which can be treated in the same way as behavioral data. This is particularly relevant because in some regression models the authors included multiple interaction terms. Was the number of observations sufficient for such a large number of predictors?

2. The authors simulated multiple agents, who behaved differently. How quickly and clearly did the participants figure out such difference? Whether and how did the process of learning the agents' preferences influence the two forms of conformity? 

3. Related to Point 2, I wonder if it was possible that the susceptible agent was treated less seriously by the participants, regardless of the participants' confidence of their own performance. This effect, if existed, would be different from the Confidence * Influence interaction that the authors considered. The assumption underlying the latter interaction (Confidence * Influence) was that the participants saw both agents as equally informative and would follow them more if the participants were less confident of their own performance. However, the interaction effect I pointed out relies on a complementary hypothesis, namely the participants treat the susceptible agent as less valuable, less independent a source of information, therefore conform to their performance less, even when the participants are not confident of their own performance. 

4. If I understand the model correctly, the model seems to only account for the agent's conformity tendency in the most recent trial. How stochastic was the agent's conformity tendency? Would it be useful to take into account more trials into the agent's history? 

5. In this task, "normative" conformity was conceptualized as an obligation to reciprocate an agent's conforming behaviors. Were the participants explicitly told that their partners would see how much they (i.e., the participants) revised their performance? This public versus private dimension matters a lot in interpreting the results of the current study (see my related comment in the next point). This critical detail was not clear to me from the authors' description. 

6. Related to Point 5, I am not sure it is appropriate to label the normative conformity in this study as "normative". Did the participants perceive the obligation to reciprocate an agent's conforming behaviors as a norm? To me, this obligation would be more accurately described as some form of interpersonal loyalty or indebtedness. There was no group norm to conform to, nor was there social pressure to enforce it (as in Asch's classic design). It seems to me necessary to at least discuss the distinction between the cognitive processes underlying conformity to a group versus imitating one single model/exemplar. Another relevant dimension is private versus public revision of behaviors, which would have critical impact on the dissociation between normative versus informational conformity. Correctly situating the mode of social context in the current study in these various forms of social influence/conformity will help the readers interpret the findings and their significance. A broader comment is that, the authors could use more self-reported data to validate their operationalization and strengthen some of their behavioral and neural results (e.g., Did the participants willingly or reluctantly revise their performance according to the performance of the susceptible agent? Did the participants accurately discern the difference between the susceptible and the insusceptible agents? Did the participants view the susceptible and insusceptible agents as equally informative?). 

7. Figure 5D shows an interesting interactive effect of confidence and influence on TPJ-dACC connectivity: when confidence was low, the effect of influence had a strong positive effect on the connectivity; however, when confidence was high, influence seemed to have a negative effect on the connectivity (i.e., not only weaker positive effect, but actually reverse). Did the behavioral interaction (Figure 2) exhibited a similar pattern? It would be helpful to illustrate the behavioral effect using the heatmap (for easier comparison with the neural effect). If, indeed, the behavioral data shows a similar pattern of interaction, I wonder how the authors would interpret the reversal. Perhaps when the participants were extremely confidence of their own performance, they would devalue the performance of the agent who always follow them (i.e., more influenced by them). 

8. The task the authors developed did not rely on binary choice. Nevertheless, it would still be useful to discuss the recent advances in the understanding of the cognitive basis of social influence afforded by studies adopting speeded binary choice tasks and drift-diffusion modeling. For example, there was a paper showing how personal and social information are integrated into decision making about rewards (Tump et al., 2020, DOI: 10.1126/sciadv.abb0266), which seems to me particularly relevant to the current study. Other examples are: Germar et al., 2013, DOI: 10.1177/0146167213508985; Son et al., 2019, https://doi.org/10.1038/s41598-019-48050-2; Yu et al., 2021, https://doi.org/10.1016/j.cognition.2021.104641

Reviewer #2: Mahmoodi et al. investigated the neural basis of social and non-social change of mind. In doing so, they used the same task which they developed previously and beautifully demonstrated both informational (i.e., effect of confidence = non-social change of mind) and normative (effect of a normative reciprocity effect = social change of mind) influence. They also found that the ACC was a key region for both types of influence and further showed that, the connectivity between the dACC and two social brain regions (namely dmPFC and PJ, both of which tracked the degree of normative influence) increased when interacting with a human partner. I think the question is very interesting, the task was creative and appropriate for testing their hypotheses, and they found results consistent with their hypotheses. However, I have several concerns which should be addressed before I recommend the paper for publication. 

Major

1. How was the fMRI sample size (n=20) determined? In general, n=20 is not considered to be many for any fMRI study with the current standard. Also, the present study used a 1.5 T scanner, which gives a lower S/N ratio than a 3.0 T scanner typically used in the field.

2. The main findings of the present study are based on the ROI analysis. The authors used a very specific ROI (their dACC ROI was exactly the same cluster reported in the previous study by Fleming et al. [2018]). The authors stated it is because the dACC "has consistently been linked to changes of mind in social (8) and non-social situations (1)." However, the dACC regions reported in refs 1 & 8 are different within the dACC (the dACC region related to social conformity is more anterior than the dACC region related to confidence). Furthermore, this strict ROI approach could lead us to misleading results. For example, although the current study showed that the same dACC area is sensitive to both social and non-social change of mind, it might well be that different regions within the dACC are associated with each of social and non-social changes of mind (and maybe overlaps in between).

3. Related to the point above, the strict ROI approach could also increase the chance of finding false-positive results if there is a researcher's degree of freedom in terms of the way of selecting a ROI. I am not saying that the authors did cherry-picking, but since this study was not pre-registered, and there are many other seemingly valid ways of defining a specific dACC ROI (e.g., use a different study that reported dACC activity related to confidence, use a study that reported social conformity related dACC activity, use Neurosynth, etc.), it is important to demonstrate how robust the present finding is especially given the small fMRI sample size. For example, can you find dACC activation (related to confidence or interaction between confidence and influence) within more broadly defined dACC ROI (e.g., anatomically defined)?

4. The same point applies to the TPJ and dmPFC ROIs.

5. Regarding the dACC-TPJ/dmPFC connectivity, it is also important to demonstrate the difference in the connectivity between the human vs. computer conditions. 

Minor:

6. As long as I understand, a total of 60 participants took part in the study, and 1/3of them performed the task inside the scanner. So, the fMRI data is based on 20 participants. I wonder if the data from all 60 participants were included in the behavioral analysis?

7. What instruction did the participants receive about the computer condition? Did they know the computer's decision was more or less accurate (thus informative)?

8. L383: It says "The first stimulus was presented for 30 ms." But, the figure 1a says it was presented for 25ms. Please clarify.

Reviewer #3: The manuscript "Distinct neurocomputational mechanisms support informational and normative Conformity" continues a line of very interesting findings probing neurocomputational mechanisms of informational and normative conformity. In brief, the results suggest that the dACC tracks informational conformity towards both human and computer but it tracks normative conformity only when interacting with human. In addition, the dmPFC and TP track normative conformity, they are also functionally coupled with the dACC when interacting with humans. The findings are interesting, but authors have to clarify some details of the study: 

1) Line 117 states that twenty participants participated in the fMRI study (N = 20) and forty participants participated in a behavioral study (N = 40).

(a) What was the reason to have a relatively small sample size in the fMRI study?

(b) From the manuscript it not clear whether results of the behavioral study are included into the data analysis, one way or another. 

(c) Did authors find a difference between "in-scanner" and "out of the scanner" behavior?

2) How did the cover story exactly explain the computer choices? I.e. how did it explain computers' revisions (e.g. the algorithm and the goal of such revisions)?

3) Line 425. From "Constructing computer partners" paragraph I got an impression that computer partners' choices differed from human partners' choices. At least, authors stated a different algorithm for the computer partners. Can you give a reason for such differences?

4) Can it be that in the observational trials participants simply lowered attention toward the screen? Such lack of attention perhaps could explain differential fMRI results for the revision and observation trials.

5) When making inferences about more than one parameter, you may use multiple comparison procedures to make inferences about the parameters of interest. Could you please comment the multiple comparison issue in the current study?

6) Authors treat the "normative reciprocity effect" as the signature of normative conformity. But according to other studies (e.g. Samuel J. Gershman, Hillard Thomas Pouncy, Hyowon Gweon, 2017) this behavioral phenomena can be understood as a form of informational social influence. To me it is still not clear if the learning of the structure of social influence represents normative conformity.

7) Line 363 suggests that "humans can and do interact with non-human social partners via informational but not normative conformity". The study address only one form of conformity - normative reciprocity effect. Will the results be replicated for other types of norms?

8) The text completely missing limitations of the study! Authors have to address this issue.

---

## [Editor Report · Decision Letter 2]

5 Jan 2022

Dear Dr Mahmoodi,

Thank you for submitting your revised Research Article entitled "Distinct neurocomputational mechanisms support informational and normative conformity" for publication in PLOS Biology. I have now discussed your new version with other staff editors and with the Academic Editor. I am pleased to let you know that we are positive about your manuscript. However, before we can make a final decision about publication, we would like you to address the following points raised by the Academic Editor:

1) Reviewer 2 (point 5) asked to see the human vs. computer interaction of the PPI effect. Instead, you report that the effects in the computer condition are non-significant, whereas the human effect is significant. This is not the same as testing the interaction (https://www.nature.com/articles/nn.2886).

2) The small fMRI sample size should be mentioned as limitation.

3) Line 618: “44 volumes of 2 mm slices”. “Volumes” is usually used to describe 3D images collected during one TR. Do you mean “slices” instead of “volumes” (i.e., “44 slices, 2 mm thick”)?

4) To avoid confusion, the text provided in the supplementary material (“Each scan run (4 in total) lasted on average 15 minutes (range: [12 to 17 min]) – generating 120 trials of behavioural data and 176-251 brain volumes for fMRI data. ”) should be part of the methods subsection “MRI data acquisition”.

Please also make sure to address the following data and other policy-related requests.

5) Title: We would like to suggest a title that might be more appealing to a broad readership. We recommend:

"Distinct neurocomputational mechanisms support informational and social conformity."

or

"Distinct neurocomputational mechanisms support informational and socially normative conformity."

However, we would be happy to work with you on an alternative if you think our suggestions misrepresent your findings.

6) Blurb: Please provide a blurb which (if accepted) will be included in our weekly and monthly Electronic Table of Contents, sent out to readers of PLOS Biology, and may be used to promote your article in social media. The blurb should be about 30-40 words long and is subject to editorial changes. It should, without exaggeration, entice people to read your manuscript. It should not be redundant with the title and should not contain acronyms or abbreviations. For examples, view our author guidelines: https://journals.plos.org/plosbiology/s/revising-your-manuscript#loc-blurb

7) Ethics: 

7.1) Please include the ID number of your protocol approved by the ethics committee at the University College London.

7.2) Please indicate if your study was conducted according to the principles expressed in the Declaration of Helsinki or any other specific national or international ethical guidelines.

8) Data: You may be aware of the PLOS Data Policy, which requires that all data be made available without restriction: http://journals.plos.org/plosbiology/s/data-availability. For more information, please also see this editorial: http://dx.doi.org/10.1371/journal.pbio.1001797

Note that we do not require all raw data. Rather, we ask for all individual quantitative observations that underlie the data summarized in the figures and results of your paper. For an example see here: http://www.plosbiology.org/article/info%3Adoi%2F10.1371%2Fjournal.pbio.1001908#s5

These data can be made available in one of the following forms:

8.I) Supplementary files (e.g., excel). Please ensure that all data files are uploaded as 'Supporting Information' and are invariably referred to (in the manuscript, figure legends, and the Description field when uploading your files) using the following format verbatim: S1 Data, S2 Data, etc. Multiple panels of a single or even several figures can be included as multiple sheets in one excel file that is saved using exactly the following convention: S1_Data.xlsx (using an underscore).

8.II) Deposition in a publicly available repository. Please also provide the accession code or a reviewer link so that we may view your data before publication. Please include a README file that explain how your data were analyzed to generate the plots of the figures mentioned below.

Regardless of the method selected, please ensure that you provide the individual numerical values that underlie the summary data displayed in the following figure panels: Figures 2B-E, 3, 4, 5AB, S1, S2, S3, S4, S5, S6, S7, S8, and S9.

8.1) Please also ensure that each figure legend in your manuscript includes information on where the underlying data can be found and that your supplemental data file/s has/have a legend.

8.2) Please ensure that your Data Statement in the submission system accurately describes where your data can be found.

We expect to receive your revised manuscript within two weeks. 

*Published Peer Review History*

*Early Version*

Sincerely,

Gabriel Gasque, Ph.D.,

Senior Editor,

ggasque@plos.org,

PLOS Biology

---

## [Editor Report · Decision Letter 3]

2 Feb 2022

Dear Dr Mahmoodi,

On behalf of my colleagues and the Academic Editor, Thorsten Kahnt, I am pleased to say that we can in principle accept your Research Article "Distinct neurocomputational mechanisms support informational and socially normative conformity" for publication in PLOS Biology, provided you address any remaining formatting and reporting issues. These will be detailed in an email that will follow this letter and that you will usually receive within 2-3 business days, during which time no action is required from you. Please note that we will not be able to formally accept your manuscript and schedule it for publication until you have any requested changes.

PRESS

Sincerely, 

Gabriel Gasque, Ph.D. 

Senior Editor 

PLOS Biology

ggasque@plos.org